# Six-Minute Walk Distance in Breast Cancer Survivors—A Systematic Review with Meta-Analysis

**DOI:** 10.3390/ijerph18052591

**Published:** 2021-03-05

**Authors:** Jasna But-Hadzic, Mirza Dervisevic, Damir Karpljuk, Mateja Videmsek, Edvin Dervisevic, Armin Paravlic, Vedran Hadzic, Katja Tomazin

**Affiliations:** 1Department of Radiation Oncology, Institute of Oncology, 1000 Ljubljana, Slovenia; jbut@onko-i.si; 2Faculty of Medicine, University of Ljubljana, 1000 Ljubljana, Slovenia; 3Faculty of Sport, University of Ljubljana, 1000 Ljubljana, Slovenia; dervisevich@gmail.com (M.D.); damir.karpljuk@fsp.uni-lj.si (D.K.); mateja.videmsek@fsp.uni-lj.si (M.V.); edvin.dervisevic@fsp.uni-lj.si (E.D.); Armin.Paravlic@fsp.uni-lj.si (A.P.); Vedran.Hadzic@fsp.uni-lj.si (V.H.); 4Science and Research Centre, Institute of Kinesiology Research, 6000 Koper, Slovenia

**Keywords:** female, cardio-respiratory fitness, reference values, breast cancer, six minute walk test

## Abstract

The six-minute walk test (6MWT) is a widely used test for the indirect measurement of cardiorespiratory fitness in various cancer populations. Although the 6MWT is a simple test, there are no normative values for breast cancer survivors (BCS) or comparisons of results with healthy counterparts. A systematic review with a meta-analysis was carried out, which included studies from 2007 to 2020. Ninety-one studies were found, 21 of which were included in the quantitative synthesis. Among them were 9 randomized controlled trials (RCT), 8 prospective cohort studies and 4 cross-sectional studies. A total of 1084 BCS were included. Our results revealed that healthy subjects (n = 878) covered a significantly greater distance than BCS during the 6MWT (589.9 m vs. 477.4 m, *p* < 0.001), and the results of the meta-regression analysis showed that the 6MWD was predicted by the participants’ BMI (*p* < 0.001), but not by their age (*p* = 0.070). After adjustment for BMI, the healthy subjects also covered greater distances than the BCS (103 m; *p* < 0.001). The normative values of 6MWT were presented for BCS. Besides, 6MWT distances distinguish between their healthy counterparts, therefore, the 6MWT distance is a relevant parameter for the assessment and monitoring of cardiorespiratory fitness in medical and exercise interventions for BCS.

## 1. Introduction

Breast cancer is the most common cancer in women worldwide. It accounted for 25.4% of the total number of new cases diagnosed in 2018 and 12.3% of all new cancer cases worldwide, regardless of gender, sharing the leading position only with lung cancer [1]. Overall, the relative 5-year survival rate for all SEER (Surveillance, Epidemiology, and End Results) stages of combined breast cancer has risen to 90% [2], due to early diagnosis and improvements in multimodal treatment over the last two decades. Accordingly, breast cancer is no longer fatal but becomes chronic as more and more people live with the side effects of breast cancer treatment (loss of muscle mass and strength, loss of mobility and upper extremity disability, lymphedema, fatigue, and cardiac toxicity) [3,4]. Indeed, multimodal treatment may leave breast cancer survivors (BCS) vulnerable to the late onset of toxic effects and at higher risk of dying from the cardio-vascular disease than from breast cancer after 65 years of age [5]. Therefore, it is necessary for BCS to engage in regular physical activity programs to maintain and/or improve their cardiorespiratory fitness (CRF) [6] and strength [7].

Although physical activity interventions in BCS are known to have a positive effect on physical fitness, quality of life, fatigue, depression, anxiety, and body composition [8], such interventions rarely meet specific guidelines, i.e., the proper frequency, intensity, duration, and type of exercise needed to improve and maintain BCS health outcomes [9]. There is no scientific evidence in the literature for prescribing and building physical activity for breast cancer survivors [10]. The latest exercise guidelines for cancer survivors suggest that an effective exercise prescription that most consistently addresses health-related outcomes experienced due to cancer diagnosis and treatment includes moderate-intensity aerobic exercise, i.e., CRF, performed at least three times a week, for at least 30 minutes and for at least 8 to 12 weeks [11]. Evidence suggests that CRF is associated with a reduced risk of dying from breast cancer in women [12] and a reduced risk of premature mortality and the incidence of CVD, respiratory disease and cancer [13]. Furthermore, an effective exercise prescription for BCS should propose a minimum intensity, duration, and frequency based on components of physical fitness and health status. Different laboratory cardio-pulmonary exercise testings (CPET) are often used for the CRF assessment. CRF is typically represented by maximal oxygen uptake (V̇O_2_max) during incremental exercise testing [14], which is commonly performed in the laboratory. Participants typically perform incremental cycling or running tests on specially designed ergometers (e.g., treadmills or cycle ergometers). These tests typically require maximal or near-maximal exertion, meaning that participants are exposed to an incremental rate of work until exhaustion at the point of maximal voluntary work rate, which is often not well tolerated in chronic patients, particularly those diagnosed with breast cancer. In addition, laboratory cardiorespiratory fitness assessment is not readily available for population testing as well as for all BCS. Therefore, several submaximal field tests have emerged as substitutes for laboratory cardiorespiratory fitness assessment. These tests are generally feasible under field conditions, inexpensive, and accurate enough to provide valid information about CRF. The most commonly used field test that indirectly measures aerobic power is the 6-minute walk test (6MWT) [15]. The 6MWT is a standardized field test used to assess functional exercise capacity and response to medical interventions in different patient populations [16,17] and to predict cardiorespiratory fitness in healthy individuals [15]. As the 6MWT is a low-cost, easy-to-perform and very well-documented field test, it is commonly used to assess walking ability in population studies [15,18] or different patient groups [19] of different ages.

The 6MWT is usually performed indoors in a 30-m corridor with two turning points and markers every 3 m [20]. The subjects are instructed to walk as far as possible in 6 minutes on a flat and hard surface, whereas a walking distance (6MWD) is the main outcome of the test. In addition, 6MWD has been shown to have an important prognostic value in patients with mild to moderate congestive heart failure, as 6MWD below 300 m was associated with higher 1-year mortality [21], while 6MWD below 468 m was associated with increased hospitalization rates in patients with stable systolic heart failure [22]. It is also a good predictor of functional endurance, treatment success, and recovery in patients with cardiac diseases [23].

The measurement properties of the 6MWT, such as reliability and validity, have been studied in patients with chronic heart failure [16], coronary artery disease [17], and cancer [24]. Specifically, the reliability of the 6MWT was evaluated in cancer patients and showed an intraclass correlation coefficient of 0.93 (95%CI: +0.86; +0.97) and a coefficient of variation of 3% [24].

In addition to good ecological validity, 6MWD showed an association with peak aerobic power, i.e., VO_2max_, in cancer patients [24] and healthy population [15]. Accordingly, it can also be recommended as a good substitute for laboratory tests in BCS patients. However, it remains unclear what is an average walking performance in 6MWT or 6MWD of BCS survivors. We can summarize that although the 6MWT is a simple and widely used test, little is currently known about the 6MWD in BCS. Knowledge of the 6MWD could be of great interest to the oncologist during regular follow-up visits to monitor the patient’s aerobic capacity and advise on exercises. In addition, this information could also be of great use to physiotherapists and kinesiologists performing exercise interventions. Tracking 6MWD could also allow for further research into the potential usefulness of 6MWD in predicting the health outcome of BCS. The purpose of this study is threefold. First, we conducted a systematic literature review with the aim of identifying the distance travelled in the 6MWT of the BCS population as a measure of their functional capacity. Second, to compare these values identified in BCS with those of the healthy female population, and finally, to investigate possible moderators of the distance travelled in the 6MWT in the BCS population.

## 2. Materials and Methods

### 2.1. Search Strategy

The procedure for this research was conducted in accordance with the Preferred Reporting Items guidelines for Systematic Reviews and Meta-Analyses (PRISMA). A member of the research team (VH) conducted a systematic search in March 2020 using three databases, including PubMed, Web of Science, and Ovid Medline, and the search terms (“six-minute walking test” OR “6MWT” OR “six minute walking” OR “6MWD” AND (“breast cancer”). No publication date range was used and no filters were applied. The articles were first searched for titles and abstracts, then duplicates were removed. Then the full text was sorted by selection criteria. The reference lists of the contained articles were analysed to identify other potentially relevant articles that were not included in the initial search. Ethical approval is not required for the study. The methods of the analyses and inclusion criteria were prespecified and documented in the protocol (PROSPERO database ID: CRD42020222847).

### 2.2. Eligibility Criteria and Study Selection

This review included original papers that: a) administered 6MWT in BCS; b) reported on sample size, mean, and standard deviation of the 6MWD (or provided sufficient information to calculate the mean); c) were available in full text. Abstracts, book chapters, reviews, presentations, and protocol papers were excluded from this analysis. In studies with a randomized controlled trial design (RCT), the quality of the studies included was assessed using the Physiotherapy Evidence Database (PEDro) scale.

### 2.3. Data Extraction

We extracted relevant information from papers that met our eligibility criteria, including sample size, age, weight, height, body mass index (BMI), disease duration, 6MWD, and testing course length. Where papers reported more than one 6MWD (e.g., values before and after intervention in randomized controlled trials, or more than one repetition of the 6MWT on consecutive days), only the baseline (initial) value or the first 6MWD was extracted.

In addition, we calculated the predicted 6MWD from Enright Sherill sex specific regression equations for women [6MWD = (2.11 × height cm) − (2.29 × weight kg) − (5.78 × age) + 667 m] [25].

### 2.4. Healthy Controls

To calculate the pooled mean of 6MWD in the healthy female population, we selected papers [18,25,26,27,28,29], in which 6MWD reference values were previously reported.

### 2.5. Statistical Analysis

The Comprehensive Meta-Analysis program [CMA version 2, Biostat, Englewood, NJ, USA] was used to calculate the difference in means between healthy and breast cancer survivors (Mean) and 95% confidence intervals (95%CI). As the variable synthesized was the same for all studies, the meta-analysis was performed directly utilizing raw values. Statistical heterogeneity was assessed using Q and I^2^ statistics. The I^2^ measure of inconsistency was calculated to determine the degree of statistical heterogeneity: low (25%), moderate (50%), and high (75%) statistical heterogeneity [30]. In addition, due to the degree of heterogeneity between studies, a random-effects model was used for all comparisons. Publication bias was assessed by examining the asymmetry of the funnel plot using Egger’s test, and significant bias was considered to exist when *p* < 0.10. In addition, meta-regression analyses (method of moments) were calculated to determine the possible moderator variables of distance walked in the 6MWT (e.g., subject age and BMI). If significant predictors were found, the raw data were adjusted to potential confounder variables, accordingly. In addition, a sub-analysis was performed with the same aim. Here, the variables were stratified as follows: for age and BMI, three different categories were formed respectively (age: under 50 years of age and over 50 years of age; BMI: healthy weight [18.5 to 24.9 kg/m^2^], overweight [25.0–29.9 kg/m^2^], and subjects with obesity [≥30 kg/m^2^]. The significance level was set at *p* < 0.05.

## 3. Results

### 3.1. Literature Search Results

The PRISMA diagram is shown in Figure 1. The first search identified 91 papers. After removing duplicates and removing studies that did not meet the inclusion criteria, 41 papers were further examined for full text. After reading the full text, 20 papers were excluded because they did not meet the inclusion criteria (e.g., no report of 6MWD, report of 6MWD from a mixed group of cancer patients without specific 6MWD for breast cancer, etc.). A total of 21 papers were included in the quantitative synthesis [31,32,33,34,35,36,37,38,39,40,41,42,43,44,45,46,47,48,49,50,51]. Among these papers were 9 randomized controlled trials (RCT), 8 prospective cohort studies and 4 cross-sectional studies. The quality assessment of the RCT studies in this review is given in Appendix A. The total score for PEDro was 4.5 (range 3–6).

The eligible studies that measured and reported 6MWD included 1084 breast cancer survivors (mean age 52 years, median age 52 years) (Table 1). Body mass and height were reported in 12 and 10 studies, respectively. Body mass index (BMI) was originally reported in 15/21 studies, with a pooled mean of 27.06 kg/m2 indicating overweight problems in the tested breast cancer survivors. In 11/21 studies, the test was performed on a 30-meter corridor, while 25- and 20-meter corridors were used in two studies. In six trials, the length of the corridor used for the test was not specified. Among healthy subjects, body mass and height were reported in 4/6 studies. Body mass index (BMI) was reported in 4/6 studies. In 5/6 studies, the test was performed on a 30-m corridor, while in one study a 45-m corridor was used (Table 2).

### 3.2. Results from the Meta-Analysis

The Egger’s test was performed to provide statistical evidence of funnel plot asymmetry (Figure 2), where the results indicated publication bias for all the meta-analyses (*p* < 0.10).

#### 3.2.1. The Meta-Analysis for the Whole sSample by Using Unadjusted Values

Our results revealed that healthy subjects covered a significantly greater distance than BCS during the 6MWT (589.9 m vs. 477.4 m, Q = 55.1, *p* < 0.001) (Table 3). When we calculated the predicted 6MWD from Enright Sherill sex-specific regression equations for females, the predicted distance was 548 m for BCS. Knowing that physical performance might be affected by intrinsic factors of the participants, such as their BMI and age, we performed an additional meta-regression analysis to investigate this relationship.

##### Findings from the Meta-Regression Analysis Conducted on the Whole Sample

The results of the meta-regression analysis showed that the distance covered at the 6MWT was predicted by the BMI of the participants (coefficient of estimate [CE] = −4.850, 95% −7.495– −2.206, Z = −3.595, *p* < 0.001), but not by the age (CE = −3.214, 95% −6.689–0.261, Z = −1.813, *p* = 0.070). Thus, the distance covered from the whole sample was adjusted by the subsequent BMI slope regression coefficient.

#### 3.2.2. The Meta-Analysis for the Whole Sample by Using Adjusted Values for BMI

Our results from the meta-analysis of adjusted values confirmed previous results, which showed that healthy subjects covered significantly greater distance during 6MWT, when compared to BCS (480.1 m vs. 377.4 m, Q = 43.8, *p* < 0.001) (Table 3). Thus, in the following sections only results from the unadjusted values will be presented.

#### 3.2.3. The Meta-Analysis for BCS Only

Our results revealed that that BCS covered 477.4 m on average (95%CI 454.0–500.8, *p* < 0.001), with high heterogeneity (Q = 618.4, I^2^ = 96.9, *p* < 0.001) (Table 3). Therefore, a sub-analysis was performed to examine potential moderators of the observed effect.

### 3.3. Influence of Different Moderating Variables on the Distance Covered during 6MWT in BCS

The following moderating variables were studied: participants age (under 50 years of age and over 50 years of age); and BMI (healthy weight [18.5–24.9 kg/m^2^], overweight [25.0–29.9 kg/m^2^], and subjects with obesity [≥ 30 kg/m^2^]).

#### 3.3.1. BMI of Participants

Our sub-group analysis indicated that subjects with healthy weight covered the greater distance on 6MWT (Mean = 515.5, 95%CI 423.3–459.9, *p* < 0.001), than overweight (Mean = 497.7m, 95%CI 471–524.4, *p* < 0.001) and/or subjects with obesity (Mean = 447m, 95%CI 406.3–487.7, *p* < 0.001), respectively (Table 3). However, the analysis failed to reach the level of significance (Q = 4.9, *p* = 0.084).

#### 3.3.2. Age of Participants

Our sub-group analysis showed that distance covered on 6MWT was moderated by age, where subjects over 50 years of age covered greater distance (Mean = 491.7 m, 95%CI 458.7–524.7, *p* < 0.001) than those under 50 years of age (Mean = 459.6, 95%CI 423.3–459.9, *p* < 0.001) (Table 3). However, the analysis failed to reach the level of significance (Q = 1.6, *p* = 0.200).

### 3.4. Findings from the Meta-Regression Analysis

The findings from the meta-regression analysis showed that neither of the included variables could predict distance covered in 6MWT among BCS population (BMI: CE = −5.466, 95% −13.697–2.766, Z = −1.301, *p* = 0.193) and (age: CE = −0.819, 95% −6.337–4.699, Z = −0.291, *p* = 0.771).

## 4. Discussion

To the best of our knowledge, this is the first systematic review with a meta-analysis of the reported distances for 6MWT in BCS. We have shown that the pooled mean of 6MWD in BCS is 477 m and that with 95% certainty the population mean is between 454 and 501 m (Table 3). Furthermore, we showed that the 6MWT still discriminated between healthy women and breast cancer survivors after adjusting for BMI, as the distance covered during the test was still 103 m longer in the healthy women (Table 3; *p* < 0.0001). Indeed, greater body mass increases the workload for a given amount of exercise, which likely resulted in a shorter distance walked during the 6MWT by BCS and healthy woman with a greater BMI. On the other hand, our adjusted values showed that the shorter walking distance covered by BCS was not only due to the BMI-related increase, but also to other factors (e.g., physical inactivity or impaired cardiorespiratory fitness). Such results were previously shown by Ortiz [41], who reported baseline data from an exercise intervention in 89 Latina BCS at mean age 56 years. In their study, 6MWD was 436 ± 99 m and was comparable to normative values in 80–89-year-old community-dwelling adults, demonstrating that cardiorespiratory fitness and gait-specific activities are severely impaired in BCS. In addition, Ying [47] also reported that the average 6MWD of patients with breast cancer was lower than in healthy comparison groups. Specifically, breast cancer diagnosis had a negative effect on 6MWD of −33.6 m walked (SE = 12.8 m, *p* = 0.010) [47].

When we used the pooled data to predict 6MWD using the Enright–Sherill Equation for women [25], the predicted distance was 548 m and was 71 m above the actual measured distance of 477 m (*p* < 0.001). In addition, Mascherini [38], who performed 6MWT in 43 stage IIIC (or less) breast cancer patients, showed that the initial mean 6MWD (497 m) was below the predicted normal range for age and sex. This underscores the importance of actual testing that allows us to assess cardiorespiratory fitness and training performance of BCS, which is necessary to set a realistic and achievable training goal. Furthermore, intervention studies (RCT and prospective cohorts) have shown a significant improvement of 6MWD after the training intervention with an average improvement of 40 m or 8% of baseline.

In most (15/21) of the studies, 6MWT was used as part of a comprehensive physical assessment of BCS [31,32,33,35,36,37,39,40,43,45,46,48,49,50,51] to evaluate the CRF-related effectiveness of the exercise intervention (Table 1). In the Ortiz study [41] mentioned above, the authors also reported significant combined effects of weekly exercise volume (expressed in minutes of moderate and vigorous physical activity with additional walking) reported by the International Physical Activity Questionnaire and 6MWD, whereas in another study examining 40 BCS aged 20-60 years, they did not report such an effect [44]. In this later study, 6MWD showed no significant correlation with the strength of the quadriceps (r = 0.168), the Charlson co-morbidity index (r = −0.194) or the hospital anxiety and depression scale - HADS (r = −0.123).

In addition, Siagian [42] recently investigated the relationship between global longitudinal strain (GLS) and 6MWD, a simple parameter expressing longitudinal shortening as a percentage (change in length relative to baseline length) used to evaluate left ventricular systolic dysfunction caused by cancer chemotherapy. In this study, 35 breast cancer patients (age: 45.83 ± 6.96 years) who received chemotherapy with an anthracycline regimen were examined before anthracycline therapy, after three cycles, and after six cycles. Patients underwent echocardiography after three cycles and echocardiography and 6MWT after six cycles. The results obtained showed a significant association between the decrease in GLS and the decrease in 6MWD (r = −0.335; *p* = 0.028), suggesting that in practice simple field test results can also be used to monitor the toxic effects of chemotherapy in BCS, making knowledge of 6MWD clinically important for oncologists.

Only two studies [34,47] have examined 6MWT in BCS more comprehensively and examined the factors that influence 6MWD. Ying [47] showed that age (-2.6 m walked per year older; *p* < 0.001) and body mass index (-4.2 m walked per unit of body mass index increase; *p* < 0.001) had statistically significant negative effects on 6MWD. This suggests that CRF decreases with age and with overweight and obesity. Surprisingly, when we used age as a moderating factor, our meta-analyses showed that older BCS women had greater walking ability than younger ones (although this difference was not significant; *p* = 0.2). Although age is a strong predictor of functional deterioration and there is much evidence that breast cancer treatment can accelerate ageing, it appears that the effect of ageing in BCS on their functional capacity is not necessarily negative. This may be supported by the 2014 findings of Champion and co-authors [52], who found that younger BCS (diagnosed at age 45) reported poorer functioning than older BCS (diagnosed between ages 55 and 70) in terms of body image, anxiety, sleep, marital satisfaction, and fear of recurrence, suggesting that younger BCS are at greater risk for long-term quality of life issues than survivors diagnosed at an older age.

In addition, the presence of comorbidities (−56.9 m walked; *p* < 0.001) also had statistically significant negative effects on 6MWD. The relationship between 6MWD and a patient’s fitness, psychological and physiological state, quality of life, cancer-related symptoms, and body composition was investigated in a cross-sectional study conducted by Galiano-Costillo [34] in 87 women diagnosed with stage I, II, or IIIA breast cancer (age: 48.33 ± 8.45 years). They concluded that the 6MWT can be used as a measure of the major components of overall health in women with breast cancer [34]. In detail, the authors showed that 6MWD is not influenced by cancer stage (*p* = 0.381), type of surgery (*p* = 0.678), lymphedema (*p* = 0.419), hormone therapy (*p* = 0.371), or type of adjuvant therapy (*p* = 0.645). A significant relationship was reported between 6MWD and several sub-scores of European Organization for Research and Treatment from Cancer Quality and Life Questionnaire Core (QLQ-C30), with the strongest correlation for overall health status and physical functioning being reported. However, it should be stressed that the sub-scores of the breast cancer specific QLQ questionnaire (QLQ-BR23) were not significantly correlated with 6MWD. In addition, cognitive performance measured by HADS was also reported to be negatively and significantly correlated with 6MWD, and this was true for anxiety (*p* < 0.001) and depression (*p* = 0.039). In terms of body composition, it appears that the percentage of lean body mass is correlated with 6MWD (*p* = 0.002) and with physical fitness components such as chair lifting (*p* = 0.01) and trunk dynamometry (*p* < 0.001). Surprisingly, the strongest correlation was reported with a short pain inventory, where pain intensity (*p* < 0.001) and pain interference (*p* < 0.001) were negatively correlated with 6 MWD. This suggests that BCSs that perform better at 6MWT have a better experience with pain or cope better with pain than those that perform worse. As pain management is a common clinical problem this further adds to the relevance of 6MWT for oncologists. To sum up, it seems that different factors were so far tested to examine their influence on 6MWD in BCS (e.g., age, BMI, cancer stage, comorbidities, etc.) while some other factors such as muscle strength or iron deficiency anaemia that were shown to impact 6MWD in other patient population [53] have yet to be examined. Our meta analysis did not show significant moderation of 6MWD concerning age and BMI within the BCS group. We believe that future studies should address more potential modifying factors including (inflammation status, motivation, etc.).

Finally, the results of previous studies support and justify the use of 6MWT in the BCS population to evaluate and follow up the CRF during exercise interventions. In addition, there is evidence that better performance is associated with a better quality of life and overall health-related fitness, making this simple, non-invasive field test even more interesting for use in a larger cohort, especially as modern wearable technology enables the reliable measurement of 6MWD with smartphones [54]. However, the current state of knowledge about 6MWT in BCS has several gaps.

First of all, there is no minimal clinically significant difference reported for this specific population, as there is one (25 m) in patients with coronary artery disease [55]. Such information would further increase the usefulness of the pooled mean we report in this review. In addition, analysis of the receiver operating curve would help to establish better cut-off values to distinguish either between healthy controls and BCS. Secondly, we believe that a properly conducted study is justified to establish a breast cancer-specific predictive equation for clinical use in this population. Thirdly, since 30-meter corridors are problematic in many health care facilities, a shorter corridor of 15 m should be tested, as this could increase the implementation of 6MWT in clinical practice. Finally, there should be consensus on the data reported along the 6MWD and we suggest that age, height, body mass, disease duration and Borg dyspnoea score should be readily reported to allow for future meta-analyses.

At the end, we would like to comment on the quality assessment of the RCT studies in this overview (Appendix A) with total PEDro score of 4.5 (range 3–6). Considering that in exercise intervention studies, the maximum PEDro score is 8 due to the inability of the blinding process for therapist and participants, we can say that the studies are of low to medium quality. This could be seen as one of the limitations of this review and we hope that the suggestions made above regarding the future use of 6MWT in breast cancer trials could help to improve this score.

## 5. Conclusions

A large body of evidence shows that low cardiorespiratory fitness (CRF) in breast cancer patients is a strong, independent, and modifiable risk factor for premature mortality and increases the risk of CVD [56]. Therefore, physical activity becomes increasingly important for breast cancer patients during treatment and rehabilitation. When prescribing physical activity for breast cancer survivors, all precautions should be taken into account, including the patient’s current fitness level and general health. Previous studies have already shown that the 6MWT is acceptable for assessing cardiorespiratory fitness and general health status, as it is cost-effective and well-documented for assessing functional capacity and response to medical interventions. However, little is currently known about the 6MWD in BCS. To address these issues and fill an important gap in the understanding of fitness levels in breast cancer survivors, we showed that the mean 6MWD was 477 m and the results of meta-regression analysis showed that 6MWD was predicted by participants’ BMI, whereas within the BCS group BMI was not a moderating factor.

We can conclude that the 6MWD can be used as a measure of important components of overall health in women with breast cancer [34], consequently, the results of our studies should be considered by researchers and clinicians when prescribing physical activity and monitoring BCS CRF levels with 6MWD. Moreover, the 6MWT is inexpensive, easy to administer, and generally considered submaximal [15]. Future studies are warranted to establish better cut-off values and predictive validity of the 6MWT in BCS as well as population-specific predictive equations for clinical use in this population.

## Figures and Tables

**Figure 1 ijerph-18-02591-f001:**
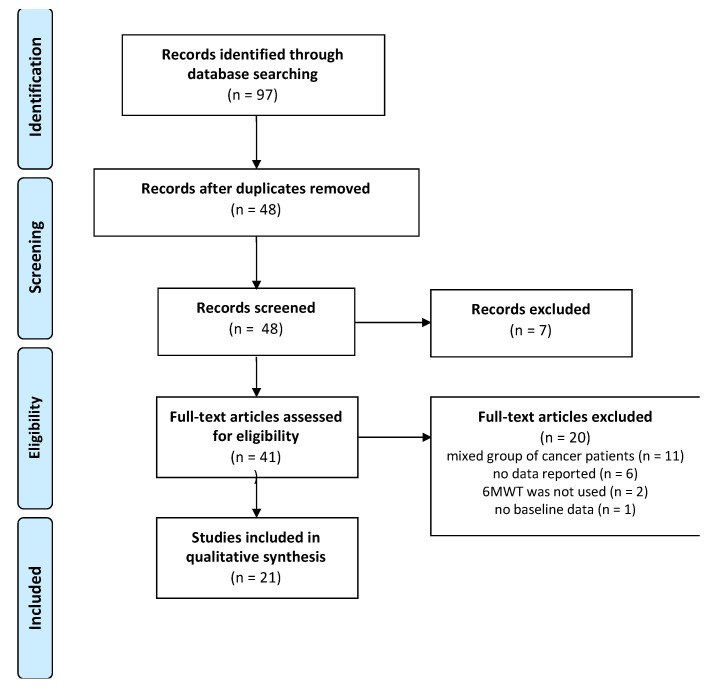
PRISMA diagram.

**Figure 2 ijerph-18-02591-f002:**
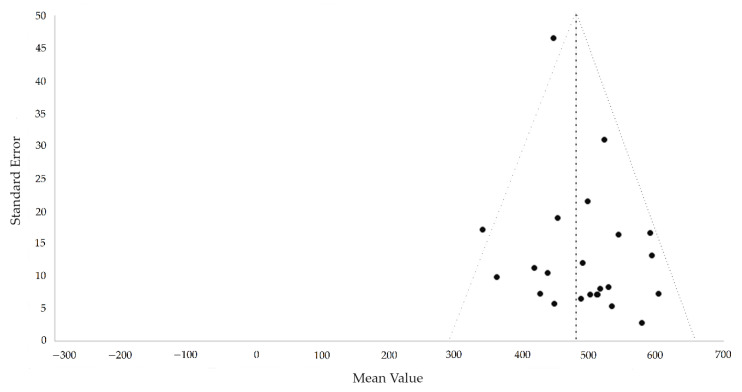
Funnel plot of the mean value vs. standard errors.

**Table 1 ijerph-18-02591-t001:** Summary of studies reporting six minute walk distance in breast cancer survivors.

Author	Study Type	Year	*N*	Age (years)	Body Height (cm)	Body Mass (kg)	BMI (kg/m^2^)	Cancer Diagnosis (years)	Corridor Length (meters)	6MWD (meters)	Cancer Stage
Hojan	RCT	2020	47	54.5 ± 5.8	166 ± 4	67 ± 4	24.8 ± 2.39	NR	30	446 ± 39	IB, IIA, IIB, IIIA
Yee	RCT	2019	14	62.2 ± 10.6	163 ± 7	75.2 ± 16.3	28.30 ± NR	9.8 ± 6.5	20	521 ± 116	IV
Cenik	cohort prospective study	2019	4	52.0 ± 2.0	NR	NR	34 ± 3	NR	NR	489 ± 24	IA, IIA, IIIA
Ortiz	cross-sectional	2018	89	55.5 ± 10.0	156.1 ± 6.2	74.7 ± 15.3	31 ± 6.5	NR	NR	436 ± 99	I - IV
Cerulli	cohort prospective study	2019	14	58.3 ± 5.2	164.8 ± 6	63 ± 5.5	23.20 ± NR	NR	NR	590 ± 62	NR
Ariza-Garcia	RCT	2019	68	48.1 ± 8.8	NR	NR	NR	NR	30	451 ± 156	I, II, IIIA
Ying	cross-sectional	2019	97	47.5 ± NR	NR	NR	27.4 ± NR	NR	30	487 ± 64	0-IV
Mascherini	cohort prospective study	2018	43	51.5 ± 9.9	161 ± 6	72.2 ± 11.2	27.9 ± 4.3	NR	NR	497 ± 141	IIIC
Siagian	cohort prospective study	2018	35	45.8 ± 7.0	NR	54.4 ± 8.3	NR	NR	30	360 ± 58	I-III
Leclerc	controlled non-randomized trial	2017	209	53.4 ± 9.4	NR	NR	25.95 ± 7.5	NR	30	533 ± 77	0, I, IIA, IIB, III
Mascherini	cohort prospective study	2017	13	49.1 ± 5.5	163 ± 7.3	70.3 ± 9.3	26.5 ± 3.6	NR	30	445 ± 168	NR
Ng	cohort prospective study	2016	21	48.0 ± NR	164 ± NR	65.4 ± NR	25 ± NR	4.3 ± NR	30	636 ± NR	0-III
Galiano-Castillo	cross-sectional	2016	87	48.3 ± 8.5	NR	NR	NR	NR	NR	339 ± 160	I, II, IIIA
Foley	cohort prospective study	2016	52	59.7 ± 10.4	164 ± 5	81 ± 2.5	30.11 ± 0.93	4.96 ± 6.3	30	417 ± 81	NR
Cornette	RCT	2016	42	52.1 ± NR	NR	NR	25.1 ± 3.3	NR	25	527 ± 53	I, IIA, IIIA, IIIB
Vardar-Yagli	cross-sectional	2015	40	48.6 ± 6.4	160.25 ± 8.9	74.4 ± 11.5	29.22 ± 5.76	4.8 ± 1.62	30	511 ± 45	I, II
Vardar-Yagli	RCT	2015	40	48.1 ± 6.2	NR	NR	29.2 ± 5.8	4.6 ± 4	30	511 ± 45	I, II
Milecki	RCT	2013	66	52.5 ± 10.8	164.1 ± 6.1	68.3 ± 14.1	25.3 ± 4.9	NR	30	425 ± 60	I - IIIC
Vincent	cohort prospective study	2013	39	49.0±8.4	NR	62 ± 12.9	24 ± 7.4	NR	25	515 ± 50	I - III
Eyigor	RCT	2010	42	49.0±8.0	NR	NR	NR	3.2 ± 2.9	20	500 ± 46	NR
Yuen	RCT	2007	22	53.9±12.8	NR	NR	NR	NR	NR	543 ± 77	NR

RCT–randomized controlled trial; BMI–body mass index, 6MWD–six minute walk distance; NR–not reported; *N*–number of participants.

**Table 2 ijerph-18-02591-t002:** Summary of studies reporting six minute walk distance in healthy women.

Author	Study Type	Year	*N*	Age (years)	Body Height (cm)	Body Mass (kg)	BMI (kg/m^2^)	Corridor Length (meters)	6MWD (meters)
Camarri	cross-sectional	2005	37	55-75	NR	NR	NR	45	628 ± 59
Chetta	cross-sectional	2006	54	33 ± 9	164 ± 7	59 ± 8	22 ± 3	30	593 ± 97
Casanova	prospective	2011	206	42-76	NR	NR	NR	30	555 ± 81
Enright	prospective	1998	173	62 ± NR	162 ± NR	NR	25.5 ± NR	30	491 ± NR
Zou	cross-sectional	2017	319	40 ± 12.0	159.0 ± 4.9	57.0 ± 7.2	22.6 ± 2.9	30	578 ± 49.9
Oiviera	prospective	2019	89	39.8 ± 13.1	162.0 ± 6.5	61.1 ± 8.5	23.3 ± 3.0	30	602.4 ± 69.0

BMI–body mass index, 6MWD–six minute walk distance; NR–not reported; *N*–number of participants.

**Table 3 ijerph-18-02591-t003:** Summary statistics for all analyses conducted.

**Independent Variables**	**Number of Studies**	**Mean**	**SE**	**95% CI**	***p***	**I^2^ (%)**	**df**	**Q value and p between Groups**
**Whole sample–unajusted values**								
Healthy	3	589.9	9.4	571.6–608.3	<0.001	80.9	2	55.1 (<0.001)
BCS	20	477.4	11.9	454.0–500.8	<0.001	69.9	19	
**Whole sample–adjusted vaues for BMI**								
Healthy	3	480.1	8.4	463.6–696.6	<0.001	84.4	2	43.8 (<0.001)
BCS	20	377.0	13.1	351.4–402.7	<0.001	98.6	19	
**BCS–age as moderating factor**								
Under 50 years of age	9	459.6	18.5	423.3–495.5	<0.001	97.2	8	1.6 (0.200)
Over 50 years of age	11	491.7	16.8	458.7–524.7	<0.001	97.0	10	
**BCS–BMI as moderating factor**								
Healthy weight	3	515.5	35.8	445.4–585.6	<0.001	98.0	2	4.9 (0.084)
Overweight	9	497.7	13.6	471.0–524.4	<0.001	95.1	8	
Subjects with obesity	3	447	20.8	406.3–487.7	<0.001	90.2	2	

SE–standard error; CI–coinfidence interval; df–degrees of freedom; *p*–value of signficance; I^2^–heterogenity; Q–value idnicating Q statistics; bold values–indicating a statistical significant results; BCS–breast cancer survivors, BMI–body mass index.

## Data Availability

The data that support the findings of this study are available from the corresponding author upon reasonable request.

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
