# Peer review of "Six-Minute Walk Distance in Breast Cancer Survivors—A Systematic Review with Meta-Analysis"

_ijerph, 2021, doi:10.3390/ijerph18052591_

Round 1

Reviewer 1 Report

This is a systematic review of studies looking at 6MWT/6MWD as a reflection of the cardiorespiratory fitness in breast cancer survivors (BCS) as compared to healthy women. It concludes that BCS have a significantly lower 6MWT than healthy women, by a mean of 88 meters. It is an interesting concept, one that can be valuable and easily implementable in clinics to measure fitness levels at baseline and after interventions and to potentially be used as a motivator for healthy lifestyle interventions. For this, the authors are to be congratulated.

I my opinion, a major flaw of this study comes from the multitude of variables contributing to the 6MWT, between the most important, recognized also by the authors, being BMI and age. The groups being studied (BCS versus healthy women) are different in respect to these 2 variables (mean age 52 for BCS versus 42 for healthy women and BMI 27 for BCS versus 23 for healthy women). Both of these variables are in favor of the healthy women, therefore we could expect a better 6MWT in healthy women by the fact that they are younger and slim are. Therefore, comparing these to groups that defer at baseline may be not appropriate.

To overcome this major limitation, I would suggest eliminating the studies with the more extreme mean BMI and mean age, in an attempt to have a more homogeneous comparison (similar BMI and similar age) or compare the 6MWT in BCS with normative values for healthy counterparts with similar mean age and BMI (if existent).

In addition, I have the following suggestions:

  • Page 1 row 37-38: the authors state that mortality in breast cancer survivors is due mainly to cardiovascular disease. From my recollection, this is true only in women after age 65. I checked the reference [5] that they refer to, but this reference is dealing with a different issue. Please find a reference to back up your statement.
  • Page 8 line 183, please correct the age to state mean age of 56.
  • The statement on page 8 between rows 186-188 is not reported in the results and not listed as an outcome. It shows up as a surprise in the discussion section. The predicted VO2 max should be listed as an outcome and the calculation explained in the methods, and its results described in the results section. Same comment for the statements between lines 195-197. Please explain the Enright Sherill equation in the methods and give these results under the results section.

Author Response

Dear Editor and reviewers,

First of all, we would like to thank you and the reviewers for the valuable comments that significantly improved our manuscript. We have carefully evaluated all your suggestions and corrections and have responded accordingly. Below are our responses to the comments. All changes made in the manuscript are highlighted in red. 

Point 1: This is a systematic review of studies looking at 6MWT/6MWD as a reflection of the cardiorespiratory fitness in breast cancer survivors (BCS) as compared to healthy women. It concludes that BCS have a significantly lower 6MWT than healthy women, by a mean of 88 meters. It is an interesting concept, one that can be valuable and easily implementable in clinics to measure fitness levels at baseline and after interventions and to potentially be used as a motivator for healthy lifestyle interventions. For this, the authors are to be congratulated.

Response 1: Thank you for your supportive comments.

Point 2: In my opinion, a major flaw of this study comes from the multitude of variables contributing to the 6MWT, between the most important, recognized also by the authors, being BMI and age. The groups being studied (BCS versus healthy women) are different in respect to these 2 variables (mean age 52 for BCS versus 42 for healthy women and BMI 27 for BCS versus 23 for healthy women). Both of these variables are in favour of the healthy women, therefore we could expect a better 6MWT in healthy women by the fact that they are younger and slim are. Therefore, comparing these to groups that defer at baseline may be not appropriate. To overcome this major limitation, I would suggest eliminating the studies with the more extreme mean BMI and mean age, in an attempt to have a more homogeneous comparison (similar BMI and similar age) or compare the 6MWT in BCS with normative values for healthy counterparts with similar mean age and BMI (if existent).

Response 2: We agree with you that a variety of variables contribute to the 6MWT, so we moderate the purpose of the study, which was changed to: "The purpose of this study is threefold. First, we conducted a systematic literature review with the aim of identifying a normative value of 6MWD in BCS. Second, to compare these values identified in BCS with those in the healthy female population. And finally, to investigate potential moderators of the distance travelled at 6MWT in the BCS population (Pgs 2-3, lns 99-103). To accomplish these goals and to address a problem with multimodality, a new statistical analysis was conducted. Therefore, the following text was added: "Comprehensive Meta-Analysis program [CMA version 2, Biostat, Englewood, NJ, USA] was used to calculate the difference in means between healthy and breast cancer survivors (Mean) and 95% confidence intervals (95%CI). As the variable synthesized was the same for all studies, meta-analysis was performed directly utilizing the raw values. Statistical heterogeneity was assessed using the Q and I2 statistics. The I2 measure of inconsistency was calculated to determine the degree of statistical heterogeneity: low (25%), moderate (50%), and high (75%) statistical heterogeneity [30]. In addition, due to the degree of heterogeneity between studies, a random-effects model was used for all comparisons. A funnel plot was used to determine potential publication bias by looking at the asymmetry of the graph. In addition, meta-regression analyses (method of moments) were calculated to determine the possible moderator variables of distance walked in the 6MWT (e.g., subject age and BMI). If significant predictors were found, the raw data were adjusted for possible confounder variables accordingly. In addition, a sub-analysis was performed with the same aim. Here, the variables were stratified as follows: for age and BMI, three different categories were formed respectively (age: up to 50 years, between 51 and 60, and 60+; BMI: healthy weight [18.5 to 24.9 kg/m2], overweight [25.0-29.9 kg/m2], and obese [≥30kg/m2]. The significance level was set at p<0.05 (Pgs 3-4, lns 141-158). Furthermore, in the results section a new data was added (Pg 4, lns 181-224 and pg 10, lns 236-238)

Point 3: Page 1 row 37-38: the authors state that mortality in breast cancer survivors is due mainly to cardiovascular disease. From my recollection, this is true only in women after age 65. I checked the reference [5] that they refer to, but this reference is dealing with a different issue. Please find a reference to back up your statement.

Response 3: Thank you for your comment. The reference has been corrected as requested and the following text added: "after 65 years of age" (Pg 1, ln 39). The reference has been changed to “Patnaik JL, Byers T, DiGuiseppi C, Dabelea D, Denberg TD. Cardiovascular disease competes with breast cancer as the leading cause of death for older females diagnosed with breast cancer: a retrospective cohort study. Breast Cancer Res. 2011; 13:1-9.” (Pg 14, lns 397-399).

Point 4: Page 8 line 183, please correct the age to state mean age of 56.

Response 4: Thank you for your comment. Mean age was corrected as requested to: “Such results were previously shown by Ortiz [43], who reported baseline data from an exercise intervention in 89 Latina BCS at mean age 56 years. In their study, 6MWD was 436±99 meters and was comparable to normative values in 80-89-year-old community-dwelling adults, demonstrating that cardiorespiratory fitness and gait-specific activities are severely impaired in BCS.” (Pg 11, ln 253)

Point 5: The statement on page 8 between rows 186-188 is not reported in the results and not listed as an outcome. It shows up as a surprise in the discussion section. The predicted VO2 max should be listed as an outcome and the calculation explained in the methods, and its results described in the results section.

Response 5: The predicted VO2max was not our main outcome as it was not possible to perform an individual calculation as the studies obtained in BCS did not report individual data. Accordingly, the following text has been removed: “When we used pooled data, e.g. 6MWD, age and body weight, to predict VO2max [15], the predicted VO2max value in BCS was 24 ml/kg/min, which is in the 30th percentile (95% CI 28% to 33%) and is consistent with reference values obtained with bicycle ergometry [14] in healthy individuals of the same age. This means that only 30% of the reference population had maximal oxygen uptake below the BCS.” Accordingly, the following text has been removed from the Conclusion section: "The predicted VO2max value from the pooled data, e.g. 6MWD, age and body weight, was 24 ml/kg/min [15], which is in the 30th percentile (95% CI 28% to 33%) when healthy women were tested [14]."

Point 6: Same comment for the statements between lines 195-197. Please explain the Enright Sherill equation in the methods and give these results under the results section.

Response 6: Enright Sherill equation has been added to the Methods section:” In addition, we have calculated the predicted 6MWD from Enright Sherill sex specific regression equations for women [6MWD = (2.11 × height cm) - (2.29 × weight kg) - (5.78 × age) + 667 m] (). (Pg 3, lns 132 - 134) In addition, following text was added in the results section: “When we calculated the predicted 6MWD from Enright Sherill sex -specific regression equations for females, the predicted distance was 548 meters for BCS.” (Pg 4, lns 132 – 134)

Reviewer 2 Report

I suggest to change the aim of the manuscript. More than a systematic review, this should be a study aimed at providing normative values. As such, there is no need for methodological quality analysis or detailed description of the studies found. The authors should provide table with age ranges, cancer stages and distance walked.

Minor comments:

I think that research in 6MWT in breast cancer is not scarce, as the authors points out. This idea should be removed. There are studies about its validity, realibility and feasibility.

The conclusion is too long.

Author Response

Response to reviewers’ comments 2

Dear Editor and reviewers,

First of all, we would like to thank you and the reviewers for the valuable comments that significantly improved our manuscript. We have carefully evaluated all your suggestions and corrections and have responded accordingly. Below are our responses to the comments. All changes made in the manuscript are highlighted in red.

Point 1: I suggest to change the aim of the manuscript. More than a systematic review, this should be a study aimed at providing normative values.

Response 1: The aim of the manuscript was changed as written below: “The purpose of this study is threefold. First, we conducted a systematic literature review with the aim of identifying a normative value of 6MWD in BCS. Second, to compare these values identified in BCS with those in the healthy female population. And finally, to investigate potential moderators of the distance travelled during 6MWT in the BCS population.” (Pgs 2-3, lns 99-103)

Furthermore, a new statistical analysis was conducted. Therefore, the following text was added: "Comprehensive Meta-Analysis program [CMA version 2, Biostat, Englewood, NJ, USA] was used to calculate the difference in means between healthy and breast cancer survivors (Mean) and 95% confidence intervals (95%CI). As the variable synthesized was the same for all studies, meta-analysis was performed directly utilizing the raw values. Statistical heterogeneity was assessed using the Q and I2 statistics. The I2 measure of inconsistency was calculated to determine the degree of statistical heterogeneity: low (25%), moderate (50%), and high (75%) statistical heterogeneity [30]. In addition, due to the degree of heterogeneity between studies, a random-effects model was used for all comparisons. A funnel plot was used to determine potential publication bias by looking at the asymmetry of the graph. In addition, meta-regression analyses (method of moments) were calculated to determine the possible moderator variables of distance walked in the 6MWT (e.g., subject age and BMI). If significant predictors were found, the raw data were adjusted for possible confounder variables accordingly. In addition, a sub-analysis was performed with the same aim. Here, the variables were stratified as follows: for age and BMI, three different categories were formed respectively (age: up to 50 years, between 51 and 60, and 60+; BMI: healthy weight [18.5 to 24.9 kg/m2], overweight [25.0-29.9 kg/m2], and obese [≥30kg/m2]. The significance level was set at p<0.05 (Pgs 3-4, lns 141-158).

In addition, in the results section a new data was added (Pg 4, lns 181-224 and pg 10, lns 236-238).

Point 2: A s such, there is no need for methodological quality analysis or detailed description of the studies found.

Response 2: Due to reviewer #1 Table 1 was removed to supplementary file.

Point 3: The authors should provide table with age ranges, cancer stages and distance walked.

Response 3: Thank you for your comment. In the available studies, the distances covered in the 6MWT were not reported as a function of cancer stages or age. In fact, there were only two studies with homogeneous subject samples according to cancer stage. Nevertheless, cancer stages were added to Table 1 (Pg 8, lns 227-231). In addition, Ying and co-workers (2019) performed a univariate analysis to determine whether individual breast cancer stage affects 6MWD. Unfortunately, the number of patients in each subcategory was insufficient to generate data with sufficient statistical significance, as indicated by the high standard errors. Therefore, a possible influence of cancer stage on 6MWD could not be demonstrated. The authors suggest that a possible future project would aim to collect 6MWT data from a population of older, less fit women without a breast cancer diagnosis and compare them to the cancer population. 

Ying, L.; Yahng, J. J.; Fisher, M.; Simons, K.; Nightingale, S., Walking the boundaries: using the 6-min walk test for accurate assessment of the level of fitness in breast clinic outpatients. ANZ J Surg 2019.

Point 4: I think that research in 6MWT in breast cancer is not scarce, as the authors points out. This idea should be removed. There are studies about its validity, reliability and feasibility.

Response 4: We agree that there are randomized controlled trials, prospective cohort studies and cross-sectional studies investigating the 6MWT in BSC. On the other hand, to our knowledge, there are no studies that have conducted a systematic review with meta-analysis that adjusted for age and BMI in BCS and compared 6MWD with that of healthy individuals.

Point 5: The conclusion is too long.

Response 5: Thank you for your comment. The conclusion was shortened.

Round 2

Reviewer 1 Report

The comments I have sent previously were addressed by performing a metanalysis of previous studies. Meta-regression methods were used for statistical analysis. I am not very familiar with these methods, therefore I can not comment on the reliability of the statistical methods used. The results seem to show the conclusion that was drawn before, that BCS have a lower 6MWD than the general population. 

Author Response

Response to reviewers’ comments 1

First of all, we would like to thank you for the valuable comments that significantly improved our manuscript. We have carefully evaluated all your suggestions and corrections and have responded accordingly. Below are our responses to the comments. All changes made in the manuscript are highlighted in red. 

Point 1: The comments I have sent previously were addressed by performing a metanalysis of previous studies. Meta-regression methods were used for statistical analysis. I am not very familiar with these methods; therefore, I can not comment on the reliability of the statistical methods used. The results seem to show the conclusion that was drawn before, that BCS have a lower 6MWD than the general population.

Response 1: We thank you for your supportive comments, which have significantly improved our manuscript. 

Reviewer 2 Report

Although the manuscript has improved, I still think that if the authors are not able to provide normative values for patients with BC, this manuscript is not worthy of being published.

They compare healthy people with women with BC. The real interest of this research would be knowing what are the mean distance that a women with BC is supossed to complete.

If the existing literature does not allow for extracting this information, then is not possible to do this, as the authors themselves say.

Author Response

Response to reviewers’ comments 2

First of all, we would like to thank you for the valuable comments that significantly improved our manuscript. We have carefully evaluated all your suggestions and corrections and have responded accordingly. Below are our responses to the comments. All changes made in the manuscript are highlighted in red.

Point 2: Although the manuscript has improved, I still think that if the authors are not able to provide normative values for patients with BC, this manuscript is not worthy of being published. They compare healthy people with women with BC. The real interest of this research would be knowing what are the mean distance that a women with BC is supposed to complete. If the existing literature does not allow for extracting this information, then is not possible to do this, as the authors themselves say.

Response: Thank you for your comments. The current systematic review with meta-analysis was conducted with three well-defined objectives, the primary of which was to identify the distance travelled in the 6MWT for the BCS population. The results have shown that the pooled mean 6MWD in BCS is 477 meters and that with 95% certainty the population mean is between 454 and 501 meters. This is copied from lines 242-244 of the submitted revised paper. Thus, this is the mean distance that women with BCS should travel on the test, and it is clear from our paper that this was our primary goal. Indeed, there are extractions in the literature, but with confounding factors that the first reviewer addressed. Therefore, we performed a meta-regression analysis to test this assumption and provided both the adjusted and unadjusted results accordingly using a meta-analysis approach. Technically, we have maintained scientific accuracy in our reporting by providing both the adjusted and unadjusted values of the 6MWD so that readers can also reach their own conclusions. We ask that you consider these comments before making your final decision.

Therefore, we have changed the aim of our study “The purpose of this study is threefold. First, we conducted a systematic literature review with the aim of identifying the distance travelled in the 6MWT of the BCS population as a measure of their functional capacity. Second, to compare these values identified in BCS with those of the healthy female population, and finally, to investigate possible moderators of the distance travelled in the 6MWT in the BCS population.” (Pgs 2-3, Lns 99-100).
